# To Err Like Human: Affective Bias-Inspired Measures for Visual Emotion Recognition Evaluation

**Chenxi Zhao** [1,2] **Jinglei Shi** [*1] **Liqiang Nie** [3] **Jufeng Yang** [1,2,4]

[1] VCIP & TMCC & DISSec, College of Computer Science, Nankai University
[2] Nankai International Advanced Research Institute (SHENZHEN· FUTIAN)
[3] Harbin Institute of Technology (SHENZHEN)
[4] Peng Cheng Laboratory
`zhaochenxi@mail.nankai.edu.cn, jinglei.shi@nankai.edu.cn`

## Abstract

Accuracy is a commonly adopted performance metric in various classification tasks, which measures the proportion of correctly classified samples among all samples. It assumes equal importance for all classes, hence equal severity for misclassifications. However, in the task of emotional classification, due to the psychological similarities between emotions, misclassifying a certain emotion into one class may be more severe than another, e.g., misclassifying 'excitement' as 'anger' apparently is more severe than as 'awe'. Albeit high meaningful for many applications, metrics capable of measuring these cases of misclassifications in visual emotion recognition tasks have yet to be explored. In this paper, based on Mikel's emotion wheel from psychology, we propose a novel approach for evaluating the performance in visual emotion recognition, which takes into account the distance on the emotion wheel between different emotions to mimic the psychological nuances of emotions. Experimental results in semi-supervised learning on emotion recognition and user study have shown that our proposed metrics is more effective than the accuracy to assess the performance and conforms to the cognitive laws of human emotions. The code is available at `https://github.com/ZhaoChenxi-nku/ECC.`

## 1 Introduction

> *"The best and most beautiful things in the world cannot be seen or even touched.*
> *They must be felt with the heart."*
> *-Helen Keller*

Emotion, as a complex state of feeling that results in physical and psychological changes and influences thoughts and behavior, involves subjective experiences, physiological arousal, cognitive appraisal and behavioral expressions [33]. Emotional similarity is a significant characteristic of emotions, revealing the commonalities among diverse emotional states in their essence, modes of expression, and influencing factors. And the essence of emotion can be explored through understanding the hierarchical nature of human cognitive processes. The Reverse Hierarchy Theory [14] in neuroscience suggests that, instead of exhibiting absolute object classification ability, humans first recognize coarse-grained categories and then proceed to identify finer-grained detailed information [21, 22]. This process involves comparing with prior information to make a comparative classification [10]. Under the same cognitive mechanism, both emotion recognition and object recognition follow a similar pattern of recognition [24], i.e., progressing from global to local, and from coarse-grained to fine-grained identification processes. And such emotional cognitive process is often characterized by the term Emotion Granularity in psychology [31, 18], where it involves the degree of nuance with which individuals can perceive. It also implies that there exists relative

---

[*]Corresponding author.

38th Conference on Neural Information Processing Systems (NeurIPS 2024).

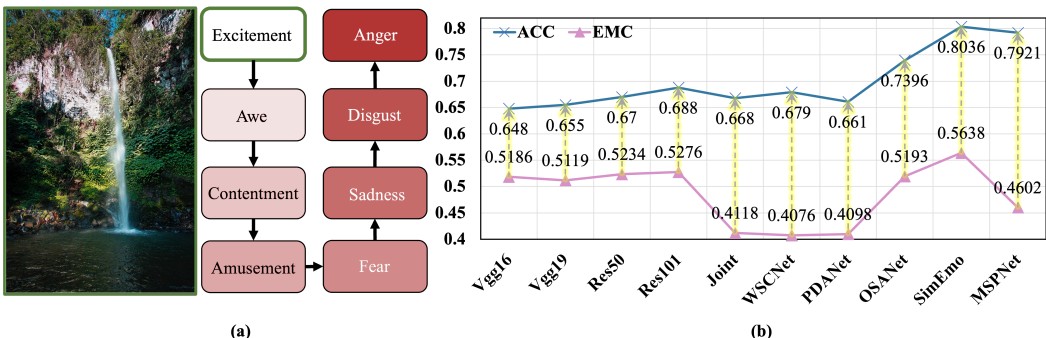

**(a)**                                  **(b)**

Figure 1: (a) Different misclassification situations can not be treated equally. It is better to misclassify an image labeled as 'excitement' as 'awe' than to misclassify it as 'anger'. (b) Although the ACC of the sentiment classification algorithm has been significantly improved on FI [50] in recent years, its Emotional Misclassification Confidence(EMC) has decreased significantly.

proximity between different emotions. For instance, belonging to positive emotions, the similarity between excitement and awe is greater than that between excitement and anger. This proximity is often associated with a misclassification-similar phenomenon in psychology referred to as 'affective biased attention', which is the tendency to pay more attention to some emotions than others. In complex psychological conditions, people tend to show negative bias compared to positive information, that is, they tend to use more negative information [36]. Though the emotion similarity and affective bias have been considered when evaluating cognitive ability for humans in psychology, the measurement of such ability for emotion recognition methods [19, 8, 25] in computer vision field still remains within the framework of the ordinary classification task. They primarily focus on employing global representation of objects [19], language prompts [8] or a multi-stage perception (entity, attribute and emotion) model [25] to improve the number of correctly classified samples, but totally ignore the influence (or severity) of different misclassifications caused by the similarity between emotions. As shown in Fig. 1(a), the relative distances between emotions are different. And after testing with the recent representative emotion classification methods in terms of both accuracy and severity of misclassification in Fig. 1(b), we observed that, although the classification accuracy has been improved to a high level, the severity of misclassification stayed at the same level or even worse. Therefore, metrics that aligns more closely with psychological models by taking the emotional similarity and the severity of misclassification into account are necessary for the visual emotion recognition task.

Studies on the misclassification for other tasks can be traced back to the early 'cost-sensitive' problem, where researchers employ the cost matrices to assess and enhance classifiers, but they care more about the class imbalance in datasets rather than the costs led by misclassification. In the field of object classification, some researchers began to pay attention to the problem of misclassification and devised methods to measure and reduce mistake severity based on the hierarchical information of labels [1, 12, 4]. For instance, Bertinetto et al. [1] devised the hierarchical distance of a mistake and average hierarchical distance of top-$K$ to quantify the severity of mistakes, and proposed the hierarchical cross-entropy loss to reduce the severity of mistakes. Garg et al. [12] proposed to learn a hierarchy-aware feature space to explicitly learn the hierarchy of labels during the training phase. All these works are based on the cost definition composed of semantic hierarchical information and lowest common ancestor, which can be traced back to WordNet [11].

However, directly applying the experience from hierarchy-based severity learning to the visual emotion recognition task is challenging, because the multi-hierarchy relationships in object categories do not exist in emotional categories, thus we can not utilize hierarchy information to define the severity of misclassification. And we could not adopt additional classifier to capture the structure available in the label space [12, 3]. While this severity is essential for designing a robust measure and effective loss function for the visual recognition task. And prior works have pointed out that making the classifier to learn both ground truth information and label structure will degrade the classification accuracy [12].

To address the above-mentioned issue, we define the concept of emotional distance (or cost) for misclassification based on the *Mikel's wheel* [23, 57], and further propose novel metrics for evalu-

ating the performance of emotion classification methods using this concept as a foundation. More specifically, we first define the cost matrix for different misclassification cases according to the definition of the distance in [57] and of the affective polarity in Mikel's Wheel. On the basis of the confusion matrix between the cost matrix and the classification results, we then propose a new visual emotion recognition evaluation measure ECC to measure all possible classification results, as well as a measure EMC focusing more on the cases of misclassification. We verify the effectiveness of the measures via the semi-supervised emotion recognition task, demonstrating that our metrics provide a more robust assessment of method performance than accuracy alone. And we prove through user study that the design of our metrics are in line with human cognition. As far as we know, both EMC and ECC are the first metrics that evaluate the performance of visual emotion classifiction algorithms by taking the influence of misclassifiction into consideration.

Our main contributions can be summarized as follows: 1). We are the first to introduce the concept of mistake cost into visual emotion recognition, and propose new measures based on Mikel's wheel to better assess the performance of emotion classification methods. 2). In semi-supervised emotion classification tasks, from the perspectives of threshold adjustment and model selection, on one hand, compared to complex confidence threshold adjustment mechanisms, our measures can be more simply yet effectively applied in the pseudo-labeling process, enhancing the model's classification performance. On the other hand, we demonstrate that our metrics can help select the model capable of generating higher-quality pseudo-labels that are beneficial for training. 3). Finally, we discussed the relationship between our metrics and $ACC_2$, and we further validate our metrics through user study, confirming that they provide results consistent with human emotional cognition in evaluating the severity of sample misclassification by the model.

## 2   Related Work

### 2.1   Visual Emotion Recognition

With the advent of deep learning, many methods [60, 30, 53, 38, 49, 29, 46, 42, 56, 54, 55] have begun to use convolutional neural network (CNN) for visual emotion recognition. Many of these researchers have explored the relationship between emotions. Based on Plutchik's [28] theory that different emotions have different similarities. Zhao et al. [57] proposed the emotional distance for the first time based on Mikel's Wheel, which is used to calculate the emotional similarity between two pairs of pictures. In addition, Yang et al. [46] designed affective polarity on Mikel's wheel and proposed a new hierarchical cross-entropy loss to distinguish between difficult and easy cases in a specific emotional way. Inspired by the large-scale pretrained language models, Deng et al. [8] propose a fine-tuning strategy based on prompts. Pan et al. [25] generate pseudo labels through visual language models as auxiliary guidance for multi-stage visual perception. However, due to the ambiguity and subjectivity of emotions, a single image often elicits multiple emotional responses, when dealing with visual emotion recognition tasks, it is more reasonable to use label distribution than single label classification. In [48], they generate sentiment distributions from a single emotion dataset based on emotional distance to solve this problem. Moveover, inspired by the inherent relationship between emotions in psychology. Yang et al. [45] propose a well-based circular structure representation to use prior knowledge to learn visual emotion distribution. However, these efforts mainly focused on improving accuracy, ignoring the fact that the severity of misclassification is not the same for different classification results, and at the same time, the constraints of fitting the label distribution may be strict, which often have the opposite effect.

### 2.2   Cost-Sensitive Classification

The importance of studying misclassification has attracted a lot of attention in the era of machine learning, but it has been neglected in the era of deep learning. In the field of machine learning, Wei et al. [40] proposed the measure of confusion entropy to evaluate the performance of classifiers, which utilizes the distribution information of misclassified categories for all categories. This problem is also described as 'cost-sensitive' by introducing the cost imbalance between different misclassifications in real-world applications, and providing solutions that meet practical needs. Classic problems include bank lending issues, disease diagnosis issues [9, 51, 4], etc. The most common cost-sensitive solution is rescaling, which mainly preprocesses the training set to improve the sensitivity to classification results. Specifically, Turney [35] studies how to choose the correct cost assignment in cost-sensitive

classification problems, and further explores the meaning and impact of cost values. Metacost [9] calculates the ideal class for each training sample by estimating the posterior probability density of the training samples and modifies the class of the original training sample to change the new data set. Different from the previous approaches, Zhou et al. [59] pointed out that in addition to the correction of the classification algorithm and dataset, the cost matrix needs to be corrected, which solves the problem that it is often ineffective for multi-class cost-sensitive learning. However, this kind of problem focuses more on the class imbalance of the dataset, and gradually loses interest in the cost of misclassification.

## 2.3 Hierarchy Aware Classification

In recent years, a small part of the work has begun to focus on the problem of the mistake severity and devised methods to reduce it. Deng et al. [6] pointed out that classification can be improved by using semantic hierarchical information from WordNet [11], which laid the foundation for future research. Furthermore, Bertinetto et al. [1] summarized and analyzed the issue of mistake severity, and proposed hierarchical cross-entropy loss to reduce the severity of mistakes. They also introduced two measures, hierarchical distance of a mistake and average hierarchical distance of top-$K$, to quantify the severity of misclassification. However, reducing the severity of misclassification is premised on reducing accuracy. Karthik et al. [17] use Conditional Risk Minimization (CRM) to improve this shortcoming, reducing the severity of mistakes without compromising accuracy. But CRM doesn't change the model, it's about making the best choice of the moment during the testing phase. In order to solve this problem elegantly, Garg et al. [12] propose to learn a Hierarchy Aware Feature(HAF) space to explicitly learn the hierarchy of labels during the training phase. In [15], they train two separate models for coarse-grained and fine-grained, and make the final prediction by calculating the normalized scores of the two models in the reasoning process. Although some progress have been made in these studies in the direction of hierarchical labeling, there is still a lack of research in visual emotion recognition. Different from they use lowest common ancestor (LCA) measure to assess the severity of mistakes, we define our measures based on Mikel's wheel and confusion matrix, and demonstrate their effectiveness.

# 3 Misclassification-Aware Measure Design

## 3.1 Definition of Emotional Distance

In almost classification tasks, accuracy serves as an important evaluation metric, measuring the model's capacity of predicting categories. However, this measure is built on a binary philosophy, i.e., only considering whether or not the predicted category matches the true label. In fact, distinguishing between different types of mismatch is meaningful, particularly in the field of emotion classification. For example, due to the similarity between emotions, misclassifying 'excitement' as 'awe' may be acceptable in certain diagnose, while misclassifying it as 'sadness' could lead to severe diagnostic errors in clinic psychology. Based on this reasoning, we therefore define emotional distances to characterize their relative relationships, and further to aid in quantifying the degree of severity of misclassification. Existing psychological models such as CES and DES, either model emotions as completely independent categories or represent them as multidimensional continuous vectors. The former ignores the similarity between emotions, while the latter requires precise measurement of emotions by experts, which can be hard to achieve for visual emotion recognition task. In [57], authors provide a definition between emotional labels based on Mikel's wheel, where the paired emotional distance is $1+$ 'wheel distance'. The 'wheel distance' means the number of steps when moving from one category to another on the wheel like shown in Fig. 2 (a). However, this definition of emotion distance neglects the emotional polarity [47], e.g., the distance between 'fear' and 'excitement' being the same as the distance between 'fear' and 'sadness'. But the distance between categories with opposite polarity should be greater than those with the same polarity, as shown in Fig. 2 (b). We thus define the emotional distance as follows:

$$W_{i,j} = \begin{cases} 1 + \text{dist}\,(e_i, e_j) & e_i, e_j \in C_p \\ \mathbf{C} + \text{dist}\,(e_i, e_j) & e_i, e_j \notin C_p \end{cases}, \tag{1}$$

where $\text{dist}\,(e_i, e_j)$ represents the number of steps on the Mikel's wheel, $\mathbf{C}$ is a constant to adapt the importance of polarity, and a larger $\mathbf{C}$ means misclassifying one emotion into the opposite polarity

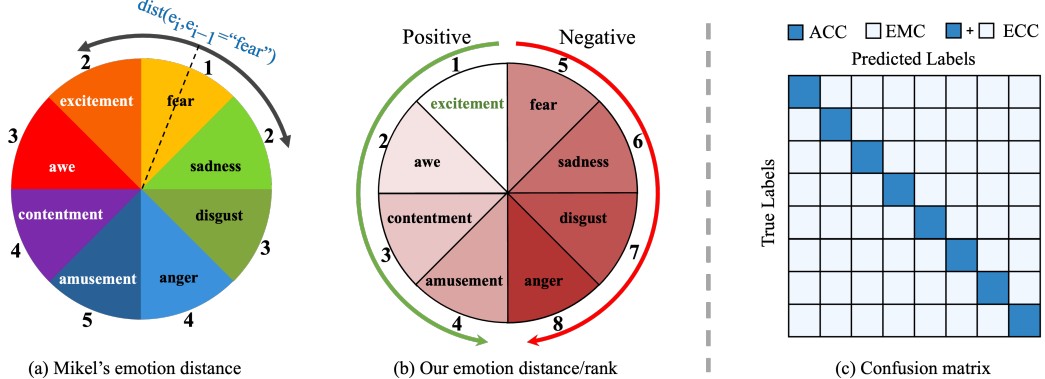

Figure 2: (a) Mikel's emotion distance. (b) Our emotion distance/rank which is labeled with 'excitement'. (c) The correspondence between the three measures of ACC, ECC and EMC in the confusion matrix.

is more severe, and we define **C** as 4 in order to ensure that the emotional distance within the same polarity is always less than the emotional distance between different polarities. $C_p$ represents the classes that have the same polarity. We can finally obtain a symmetric cost matrix with each element being the distance between the corresponding emotions. Although our emotional distance is defined based on Mikel's wheel for classifying eight emotions, it is important to note that when calculating the classification of six emotions based on Ekman's model, we can still use a similar method for determining emotional distances. To achieve it, we can initially follow the definition used by Mikel's wheel, calculating the emotional distances for the same emotions present in both models. Subsequently, we can adjust the distances between opposite emotion labels based on emotional polarity. Due to the inherent nature of the distance definition in Mikel's Wheel, our main focus here is on the issue of classifying the eight categories of emotions.

### 3.2 Emotion Confusion Confidence (ECC)

Wei et al. [40] used the category distribution information of all categories of misclassification to propose the concept of confusion entropy to measure the standard. Inspired by them, we design our measures based on a confusion matrix. As shown in Fig. 2 (c), the confusion matrix is defined as a N×N matrix, where N represents the number of classes. The rows of the confusion matrix represent the true labels, while the columns represent the predicted ones. Each element in $S_{i,j}$ represents the number of samples classified from the correct category $i$ to the category $j$. The diagonal elements of the confusion matrix represents the number of samples correctly classified for each class. Hence the sum of diagonal elements of confusion matrix $N_c$ divided by the total number of samples N represents ACC. While other elements represent the number of samples belonging to the category $i$ that are wrongly classified into category $j$. The drawback of accuracy lies in its exclusive consideration of correct classifications along the diagonal of the confusion matrix, while disregarding equally important misclassifications elsewhere in the matrix. This implies that accuracy may be misleadingly high in scenarios of class imbalance. Although directly using the confusion matrix as a metric allows for the consideration of both correct and incorrect classifications (misclassifications), this metric overlooks the distance between emotions in psychology. To take both correct classifications, misclassifications and emotional distance into evaluation, we propose to use emotional distance to modulate the confusion matrix. Based on this design philosophy, the ACC can be re-formulated as the product of the confusion matrix and a modulation factor $M_{i,j}$ as:

$$\text{ACC} = \frac{N_c}{N} = \frac{\sum_{j=1}^{c} \sum_{i=1}^{c} S_{i,j} \times M_{i,j}}{N}, M_{i,j} = \begin{cases} 1 & i = j \\ 0 & i \neq j \end{cases}, \tag{2}$$

where $c$ denotes the number of all classes, $N_c$ represents the count of correct classifications. It implies that one sample is counted as 1 only if it is correctly classified, other misclassified samples are not distinguished and are all recorded as 0. However, as we explained previously, misclassifications can be acceptable to some extent when the emotional classes are similar. Thus we tackle these samples

as 'quasi correctly classified samples' and weighted with a value between $(0, 1)$. The smaller the similarity of emotions, the less acceptable the misclassified sample becomes, hence the smaller the corresponding value it will have, and vice versa. To achieve this, we rely on the reciprocal of the emotional distance $W_{i,j}$ defined in Eq. 1 to replace the modulation factor $M_{i,j}$ in Eq. 2 for the misclassified samples to obtain ECC in Eq. 3.

$$\text{ECC} = \frac{\sum_{j=1}^{c} \sum_{i=1}^{c} S_{i,j} \times \frac{1}{W_{i,j}}}{N} = \text{ACC} + \frac{\sum_{j=1}^{c} \sum_{i=1,i\neq j}^{c} S_{i,j} \times \frac{1}{W_{i,j}}}{N} \tag{3}$$

In this way, misclassified samples are no longer simply neglected like in ACC, but have weights in terms of emotional distances, which also means that our measure ECC make use of the entire confusion matrix, including elements both inside and outside the diagonal as shown in Fig. 2(c).

### 3.3 Emotional Misclassification Confidence (EMC)

In some practical scenarios, people pay more attention to cases of misclassification, like estimating the severity of misdiagnosis of a certain mental diease, while both ACC and ECC include the cases of correct classification, hence failing to provide information about misclassification. Therefore, in order to only consider the cases of misclassification, one can extract the term $(\sum_{j=1}^{c} \sum_{i=1,i\neq j}^{c} S_{i,j} \times \frac{1}{W_{i,j}})/N$ that excludes ACC from Eq. 3 as an indicator for evaluating misclassification. However, directly using this terms as a misclassification measure is inappropriate for two reasons: 1). The denominator of this term is N, indicating that its value is still influenced by correct classified samples, rather than solely considering the misclassified samples. 2). The maximal value of this term is 0.5, which we hope to be 1 as ACC and ECC. We accordingly modify this term and propose a novel measure for misclassification description in Eq. 4, named Emotional Misclassification Confidence (EMC):

$$\text{EMC} = \frac{\sum_{j=1}^{c} \sum_{i=1,i\neq j}^{c} S_{i,j} \times \frac{1}{W_{i,j}-1}}{N - N_c} \tag{4}$$

This metric considers only misclassifed samples $S_{i,j}, i \neq j$ and their number $N - N_c$, meanwhile modifying the modulation factor from $\frac{1}{W_{i,j}}$ to $\frac{1}{W_{i,j}-1}$ to ensure a maximum value of 1.

In this way, we have an ECC that measures the overall classification results, as well as EMC that is specially designed to measure the severity of misclassification. The cooperation of two measures can better evaluate the visual emotion recognition task.

## 4 Experiments and Results

### 4.1 Datasets

We evaluate our metrics on two widely applied datasets EmoSet [44] and FI [50]. EmoSet is based on Mikel's eight-categorical sentiment model, which uses 810 keywords and collects from four different sources, including openverse, pexels, pixabay and rawpixels. It covers different emotional attributes, i.e., low-level (brightness and colorfulness), mid-level (scene type and object class), and high-level (facial expression and human action). Finally, 60 annotators who passed the test annotated a total of 118,102 images. The FI dataset was collected from Flickr and Instagram through eight sentiment keywords, and was built based on Mikel's eight-category sentiment model, which contains about 23,308 images.

### 4.2 Applications to Semi-Supervised Emotion Recognition

Because of the ambiguity of emotions [16], annotating high-quality and large-scale datasets for visual emotion recognition is arduous and challenging. Semi-supervised learning is an effective solution which consists of training the model based on both labeled and unlabeled data, then annotating unlabled examples by this trained model. While for pesudo labeling based semi-supervised learning methods, selecting appropriate models and labeling confidence thresholds play key roles in determining the final performance. In this section, we carry out experiments on two perspectives: the model selection and the adjustment of labeling threshold, to demonstrate how our proposed measures can benefit this task.

Table 1: We conduct experiments on the datasets FI [50] and EmoSet [44], and evaluate the experimental results using ACC. It is considered fair to compare our method in 4.2.1 with FixMatch and FlexMatch, given that all of them employ threshold adjustment methodologies.Where 'TA' means threshold adjustment.Based on $S^2$-VER, we compared our method in 4.2.2 with all the state-of-the-art semi-supervised methods.

| | | FI | | | EmoSet | | |
|---|---|---|---|---|---|---|---|
| | label num | 80 | 800 | 1600 | 400 | 4000 | 8000 |
| TA | Fixmatch [32] | 28.2±0.78 | 37.4±0.51 | 42.2±0.29 | 31.1±0.41 | 42.3±0.65 | 45.8±1.25 |
| | Flexmatch [52] | 29.7±0.90 | 38.2±0.49 | 40.6±0.55 | 30.4±0.78 | 42.8±0.34 | 44.9±1.24 |
| | Ours( 4.2.1) | **31.2±0.12** | **40.8±0.34** | **42.7±0.21** | **31.6±0.56** | **43.7±0.69** | **47.6±0.61** |
| State-of-the-art | Comatch [20] | 36.7±0.87 | 43.5±0.39 | 47.9±0.26 | 30.3±0.97 | 44.2±0.41 | 46.8±0.49 |
| | Simmatch [58] | 31.4±1.26 | 41.9±0.57 | 43.7±0.62 | 36.3±0.22 | 44.7±0.34 | 50.2±0.71 |
| | Freematch [39] | 26.0±1.66 | 37.3±0.43 | 39.9±0.87 | 31.2±1.63 | 41.5±0.59 | 46.3±0.61 |
| | Softmatch [5] | 30.7±1.31 | 37.9±0.78 | 40.7±0.19 | 30.8±0.35 | 44.0±1.28 | 45.8±0.25 |
| | $S^2$-VER [16] | 39.1±0.66 | 46.9±0.46 | 51.8±0.21 | 44.9±0.35 | 57.5±0.51 | 60.2±0.34 |
| | Ours( 4.2.2) | **40.2±1.08** | **48.9±0.91** | **52.1±0.33** | **47.0±0.18** | **59.0±0.33** | **61.5±0.12** |

### 4.2.1 Adjustment of Confidence Threshold

For pseudo label-based semi-supervised learning methods, a confidence threshold is used to determine whether current labeled samples are filtered will directly affect the number of training samples and the proportion of correct labels in each epoch: a high threshold will excessively filter out high-quality unlabeled data, leading to insufficient training data, while a low threshold will allow more low-quality unlabeled data, making the model converge to a poor local minima. How to select an appropriate threshold of pseudo-labeling in the training process has always been a hotspot for semi-supervised learning [5]. Existing methods either use an intuitive constant threshold [32] that is widely adopted in other semi-supervised learning tasks, or a dynamic threshold that varies in terms of the estimated learning status [52]. Considering that EMC can measure the severity of misclassification, we therefore recommend using EMC to dynamically adjust the confidence threshold before each epoch. This is similar to how, in a clinical scenario, a patient not only refers to the doctor's current confidence in the diagnosis but also takes into account the doctor's historical reputation (frequency of misdiagnoses and medical errors) to ultimately judge the reliability of the current diagnosis. For semi-supervised emotion recognition tasks, since there are only a few labeled samples available for training, the model can easily overfit these samples, resulting in all correct predictions and thus making it impossible to calculate EMC. To address this situation without introducing additional training data or increasing computational payload, we treat the pseudo-labels generated from weakly augmented unlabeled samples as the ground truth, and use the predicted labels from strongly augmented samples as the model's predictions to calculate EMC. Since the labels of the same sample should remain consistent under different data augmentation methods, a large EMC indicates that the model provide similar emotional predictions with different augment methods. This suggests that the model has grasped the underlying visual elements that represent emotions in images, which reflects the reliability and high quality of the pseudo labels. Therefore, we can lower the confidence threshold, allowing more pseudo-labeled samples to participate in the training process. On the contrary, when the EMC is small, we can raise the threshold and filter out the low-quality samples. We can realize the above confidence adjustment mechanism by simply setting:

$$\tau'_t = \tau \cdot \frac{e}{\mathbf{EMC}_t}, \tag{5}$$

where $\tau$ is the pre-defined threshold, $\tau'_t$ is the new threshold at time step $t$, $\mathbf{EMC}_t$ represents the EMC at time step $t$, $e$ is a constant for different datasets. We define $\tau$ as 0.95. For FI, we define $e$ as 0.5, and for EmoSet, we define $e$ as 0.4. And we set the upper and lower bounds of $\tau'_t$ to 0.98 and 0.7 [16] respectively to ensure stability of the training process.

To test the effectiveness of our proposed confidence adjustment mechanism built on EMC, we compared it with two representative pesudo labeling based methods, Fixmatch [32] and Flexmatch [52],

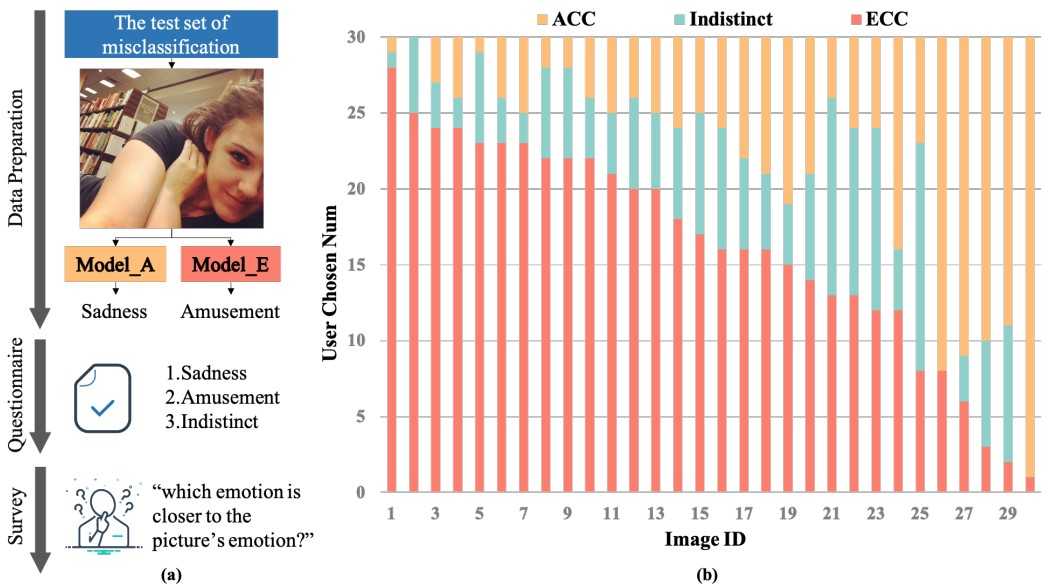

Figure 3: User study about our measures. (a) The pipeline of our user study. (b) The result of user study, where the horizontal axis is the id of the image, the vertical axis is the number of votes, and the results of different options have been distinguished by different colors.

where Fixmatch directly provides a fixed threshold, while Flexmatch assigns dynamic thresholds to each class according to corresponding learning status. Experimental results in Tab. 1 show that, thanks to our EMC-based threshold mechanism, our method is over 1% better than Flexmatch and Fixmatch. Additionally, let us note that unlike the method Flexmatch that adjusts the threshold for every category, we evaluate the entire ublabeled data based on a statistical perspective, and adjust the overall threshold after measuring the EMC, which is computationally low-cost. And the experiment is followed by [32].

### 4.2.2 Selection of Better Pseudo Labeling Models

Existing methods that typically use ACC to measure the model's capability of discriminating between categories of unlabeled samples, i.e., models with higher ACC can provide more reliable pesudo label. However, such models often suffer from the problem of confirmation bias [34], where the model will gradually deepen this error during the learning process. Accumulation of these errors eventually leads to the final model being unable to achieve good classification performances. As both ECC and EMC are designed based on the consideration of the cases of misclassification, which means they can better distinguish the ambiguity of labels, and models selected in terms of ECC or EMC will less effected by the cumulative confirmation bias, providing high-quality labels than those based on ACC.

To prove that model with higher ECC or EMC is better for pesudo labeling, we train the same network with different loss functions: cross-entropy loss $\mathcal{L}_{CE}$ and the combination of cross-entropy loss and order-based loss ListMLE [41] as $\mathcal{L}_c = \mathcal{L}_{CE} + \alpha\mathcal{L}_{ListMLE}$, where $\alpha$ is 1 and $\mathcal{L}_{ListMLE}$ aims to constrain the final prediction probability of the samples to follow a preset order, thus favoring higher ECC and EMC (proved in Appendix A). This constraint will reduce the severity of misclassification, at the same time, the cost is to reduce ACC [1], specific experiment is in Appendix B. More precisely, we first train the model with $\mathcal{L}_{CE}$, once it starts to converge (the model has preliminary ability of recognition), we then keep the same loss function or replace $\mathcal{L}_{CE}$ with $\mathcal{L}_c$ to make it continue to focus on the correct classification or focus more on the cases of misclassification. Therefore, models trained solely using $\mathcal{L}_{CE}$ exhibit better ACC but limited capability to distinguish error samples, and the quality of the pseudo label is poor. Although the pseudo labels might still be incorrect compared to the ground truth when using a combined loss function for training the model, the pseudo-labels become closer to the ground truth. In such cases, these pseudo labels can still have a positive impact on the training process and thereby improve the model's accuracy. As we show in Tab. 1, we adopt the state-of-the-art method $S^2$-VER [16] as our baseline, as it generates more reliable pseudo labels by

Table 2: We conducted experiments on the FI [50] and EmoSet [44] datasets using three backbones. As mentioned above, the loss function employed the commonly used the cross-entropy loss $\mathcal{L}_{CE}$ and the combined loss $\mathcal{L}_c$.

| Dataset | FI | | | | | | EmoSet | | | | | |
|---|---|---|---|---|---|---|---|---|---|---|---|---|
| Backbone | Resnet18 | | Resnet50 | | Resnet101 | | Resnet18 | | Resnet50 | | Resnet101 | |
| Loss function | $\mathcal{L}_{CE}$ | $\mathcal{L}_c$ | $\mathcal{L}_{CE}$ | $\mathcal{L}_c$ | $\mathcal{L}_{CE}$ | $\mathcal{L}_c$ | $\mathcal{L}_{CE}$ | $\mathcal{L}_c$ | $\mathcal{L}_{CE}$ | $\mathcal{L}_c$ | $\mathcal{L}_{CE}$ | $\mathcal{L}_c$ |
| ACC | **65.8** | 64.4 | **67.6** | 66.2 | **68.1** | 65.6 | **73.9** | 72.4 | **76.2** | 74.3 | **76.7** | 74.5 |
| $ACC_2$ | 79.0 | **86.2** | 83.7 | **86.0** | 84.7 | **86.0** | 85.0 | **85.6** | 85.3 | **85.8** | 85.7 | **85.8** |
| ECC | **76.1** | 75.8 | **77.2** | 76.8 | **77.9** | 76.7 | **82.6** | 81.9 | **84.2** | 83.2 | **84.5** | 83.3 |
| EMC | 50.1 | **54.8** | 49.0 | **53.2** | 51.9 | **54.8** | 57.0 | **59.3** | 57.0 | **59.9** | 56.9 | **60.5** |

calculating the similarity between emotional prototypes and samples, but ignores the error of pseudo labels. We follow the experimental setting of [16] and vary the proportions of the labeled samples as 0.5%, 5% and 10% (corresponding to 80, 800, 1600 label number for FI, and 400, 4000, 8000 label number for EmoSet), it can be observed that under the different settings of the two datasets, our method performs favorably against $S^2$-VER by 1% in accuracy. Meanwhile, our method also far surpasses multiple state-of-the-art methods in semi-supervised learning. It indicates that choosing a model with better misclassification ability (better ECC and EMC) can produce pseudo labels of better quality and beneficial to the training process, thus achieving better semi-supervised performance.

## 4.3 Compare with Other Measure

$ACC_2$ is a very important binary classification metric in the field of emotion recognition [43, 8, 26, 44], used to measure whether the classification to the same emotional polarity is correct. More specifically, when a sample labeled as 'excitement' is classified as 'awe', it is incorrect to use accuracy for evaluation. However, for metrics like $ACC_2$, such a classification is considered correct. In a certain sense, metrics like $ACC_2$, which involve coarse-grained classification, take into account the proximity of labels and consider misclassified cases. This approach aligns with the objectives of our measures. To further demonstrate the effectiveness of our metrics, As mentioned above, we conducted experiments using both the cross-entropy loss $\mathcal{L}_{CE}$ and the combined loss $\mathcal{L}_c$. As shown in Tab. 2, although the accuracy of the combined loss is lower than that of the cross-entropy loss, its $ACC_2$ is higher. This indicates the shortcomings of using accuracy alone in certain situations, as it fails to measure for misclassifications. Although the ECC also decreased due to the influence of the ACC, since the ECC takes into account the situation of misclassification, the gap between the two models in terms of ECC is not significant. As EMC considers metrics for misclassification alone, the EMC of the combined loss is significantly higher than that of the cross-entropy loss. In this regard, the trend of EMC aligns with that of $ACC_2$, which also demonstrates the correlation between the two metrics. In the confusion matrix, $ACC_2$ actually represents the proportion of correctly classified samples in the top-left and bottom-right sections. This also indicates that our measures are actually more refined measures that lies between Accuracy and $ACC_2$.

## 4.4 User Study

Since our metrics is founded on principles of human cognition, we aim to further demonstrate that our measures align with human judgments in emotion classification results via user study.

**Data preparation** In order to have models having different levels of ECC, we take ResNet50 as our network backbone and train it respectively with cross-entropy loss and combined loss $\mathcal{L}_c = \mathcal{L}_{CE} + \alpha \mathcal{L}_{ListMLE}$ on the FI dataset, where $\alpha$ is 0.2. Then we perform predictions on the test set and select the images that are misclassified by both two models into different classes. Finally we randomly select 50 eligible images, and filter out the images with no obvious emotion or ambiguous emotion, getting 30 carefully selected images as our tested images for user study.

**Preference Study** We invite 30 participants having different social backgrounds to our user preference study, and the test for every participant lasts about 15 minutes. During the test session, each misclassified image will be presented to participants with three options: the incorrect class predicted

Table 3: The table of comparative analysis of the impact of label ranking on single visual emotion classification task. We used ListMLE loss to do experiments on FI. 'Our Rank' stands for the Mikel's emotion rank we defined based on Mikel's wheel.'RA' means random scrambled labels, and 'RE' means scrambled labels in reverse rank. 'w/o R1' means keeping the ground truth rank first when scrambling the Label rank. Among them, the red font represents the best.

| Label Rank | Resnet18 | | | Resnet50 | | | Resnet101 | | |
|---|---|---|---|---|---|---|---|---|---|
| | ACC | ECC | EMC | ACC | ECC | EMC | ACC | ECC | EMC |
| RE w/o R1 | 60.6 | 70.2 | 40.4 | 63.4 | 73.1 | 43.3 | 63.8 | 73.6 | 44.4 |
| RA w/o R1 | 61.2 | 71.5 | 48.6 | 63.8 | 74.1 | 48.6 | 64.4 | 74.6 | 49.5 |
| Our Rank | **63.9** | **75.1** | **58.4** | **65.7** | **77.0** | **51.5** | **67.9** | **77.6** | **52.2** |

from the model trained with cross-entropy loss, the incorrect class predicted from the model trained with the combined loss function, and a third 'Indistinct' option for cases where participants are unable to discern the emotion of the image. And participants will choose their preferred options after viewing each image.

**Results** In all 900 collected votes, 487 votes are cast for the results produced by the model with higher ECC, while 242 votes opt for the results generated by the model with higher ACC. There are 171 votes that opted for unidentifiable choices. Fig. 3 further shows the distribution of the votes for each image, where we can observe that among 30 tested images, users preferred the classification results provided by the model with higher ECC over the model with higher ACC for 24 images, representing 80% of the total tested images.

### 4.5 Validity of Emotional Distance Definitions

We want to further explore the rationality of the defined emotional distance and determine whether it can help models learn the semantic structure of labels. To answer the above question, we transform emotional distance into ranking, and designed three sets of experiments based on $\mathcal{L}_c$ and $\alpha$ is 1. The specific experiments are shown in Table 3. And the specific experimental settings are detailed in Appendix C. Since we are only focusing on the impact of emotional rank (emotional distance) on the model, and changing the rank of ground truth label would prevent the model from training on correct classification categories. So we randomly shuffle and reverse the rank of the other labels while keeping the category with the first position in the emotional rank as the ground truth. Then, according to the changed order, use ListMLE for training. As shown in Table 3. Our rank achieve advanced performance in three measures. And the results are worse in reverse rank than random rank, and they are significantly worse than our rank. This shows that our rank is better, in line with human cognition of label rank, and our label rank will help the model learn the emotion category structure. Although the ranking of ground truth has not changed, a reasonable label ranking can often reflect the emotional and visual element relationship between images, which will enable the model to mine the visual and semantic correlation between similar categories, so as to learn a better label semantic structure. We have proven the rationality of emotional distance through above experiments. Since the new measures ECC and EMC are designed based on emotional distance, it also validates the rationality of our measures.

## 5 Conclusions

In this work, we define the concept of misclassification in the field of visual emotion recognition, and propose new measures to evaluate the mistake severity in visual emotion recognition based on Mikel's Wheel distance. We define our emotional distance using the Mikel wheel and adopt it to build cost matrix, then exert it to confusion matrix to compute emotion confusion confidence (ECC) and emotional mistakes confidence (EMC). And we demonstrate that our measures are more robust in semi-supervised learning. Our measures can not only help to select the model that can produce high-quality pseudo labels, but also can be used as a reference standard to adjust the threshold adaptively. Moreover, we verify that our new measures are consistent with human emotional cognition through user study. Finally, we verify the validity of our emotional distance.

# 6 Acknowledge

This work was supported by the Natural Science Foundation of Tianjin, China(NO.20JCJQJC00020, NO.22JCQNJC01560), the National Natural Science Foundation of China (NO.62302240), Fundamental Research Funds for the Central Universities, Supercomputing Center of Nankai University (NKSC).

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

# Appendix

## A  Proof of the Relationship between New Measures and ListMLE

In this chapter, we mainly analyze the relationship between ACC and cross-entropy, ListMLE and ECC. To prove that the relationship between ListMLE and ECC is equal to the relationship between cross-entropy and ACC. so as to explain why ListMLE can be used as a backbone between ECC. First of all, let's review the formula of cross-entropy loss:

$$\mathcal{L}_{ce} = -\frac{1}{n}\sum_{i=1}^{n}\sum_{j=1}^{c} y_{ij} \log f_j(x_i;\theta), \tag{6}$$

where $n$ represents the number of samples and $c$ represents the number of categories. $y_{ij}$ represents the $j$th element of one-hot encoded label of the sample $x_i$. $\theta$ is the parameter set of the classifier. $f_j(x_i;\theta)$ represents the probability that the prediction of the $i$th sample is of category $j$.

For a single sample, the formula becomes:

$$\mathcal{L}_{ce} = -\sum_{j=1}^{c} y_j \log f_j(x;\theta) \tag{7}$$

However, the formula of ACC is:

$$\text{ACC} = \frac{\sum_{j=1}^{c}\sum_{i=1}^{c} S_{i,j} \times M_{i,j}}{N} \tag{8}$$

In fact, the effect of $y_j$ is the same as that of $M_{i,j}$. the optimization goal of cross-entropy is to maximize the prediction probability of real categories, while ACC only calculates the number of samples predicted to the correct category.

In fact, the cross-entropy loss is also sensitive to the order. According to the paper [2], the cross-entropy loss can be written in the form of likelihood loss. Suppose that $\pi$ is a permutation of n objects, and $\phi$ is a strictly increasing positive function, then the probability of permutation $\pi$ of a given score list $s$ is defined as

$$P_s(\pi) = \prod_{j=1}^{n} \frac{\phi\left(s_{\pi(j)}\right)}{\sum_{k=j}^{n} \phi\left(s_{\pi(k)}\right)} \tag{9}$$

In addition, Top One Probability is defined as:

$$P_s(j) = \sum_{\pi(1)=j,\pi\in\Omega_n} P_s(\pi) \tag{10}$$

If the predicted ranking score for a given category is given, then the cross-entropy is equal to the row that wants to put ground truth first in the ranking:

$$\mathcal{L}_{ce} = -\sum_{j=1}^{c} y_j \log f_j(x;\theta) \approx -\log P_s(j), \tag{11}$$

If we want to consider the label correlation in the following sorting function, we only need to change the permutation probability of Packers $(j)$ to the sorting expectation for all categories. If we want to consider the label correlation in the later sorting function, we only need to change the permutation probability of $P_s(j)$ to the sorting expectation for all categories.

$$\mathcal{L}_{ListMLE} = -\log P_s(\pi) \tag{12}$$

Here, we get the expectation permutation $\pi$, which is the emotional distance that we define. The transformation of the likelihood function form of cross-entropy into ListMLE form is actually the probability arrangement of prediction, from what is expected to be the first element to expecting all elements to satisfy our defined element arrangement. So in terms of formula, the difference between ECC and ACC is the difference in weight $M_{i,j}$ and $\frac{1}{W_{i,j}}$. So in terms of formula, ACC is transformed into ECC, that is, $M_{i,j}$ is replaced by emotional distance.

$$\text{ECC} = \frac{\sum_{j=1}^{c}\sum_{i=1}^{c} S_{i,j} \times \frac{1}{W_{i,j}}}{N} \tag{13}$$

Table 4: The results of experiments on three single-label classification datasets, FI, EmoSet and UnbiasedEmo, in which experiments are carried out on multiple classical baselines based on our proposed loss function method, and the results on three measures ACC, ECC and EMC are reported.

| Backbone | | Resnet18 | | | Resnet50 | | | Resnet101 | | |
|---|---|---|---|---|---|---|---|---|---|---|
| Dataset | Alpha | ACC | ECC | EMC | ACC | ECC | EMC | ACC | ECC | EMC |
| | 0 | 66.2 | 76.2 | 51.5 | 67.3 | 77.0 | 51.5 | 67.9 | 77.6 | 52.2 |
| FI | 0.2 | 65.2 | 76.3 | 55.8 | 67.3 | 77.5 | 54.9 | 67.7 | 77.8 | 55.2 |
| | 1.0 | 63.9 | 75.1 | 58.4 | 65.7 | 76.2 | 57.3 | 65.9 | 76.6 | 61.8 |
| | 0 | 73.8 | 82.4 | 57.7 | 76.3 | 84.0 | 58.5 | 76.9 | 84.5 | 58.9 |
| EmoSet | 0.2 | 73.1 | 82.5 | 60.3 | 75.5 | 84.1 | 60.1 | 75.9 | 84.3 | 59.8 |
| | 1.0 | 72.2 | 81.8 | 65.5 | 74.5 | 83.3 | 61.7 | 74.5 | 83.3 | 61.3 |

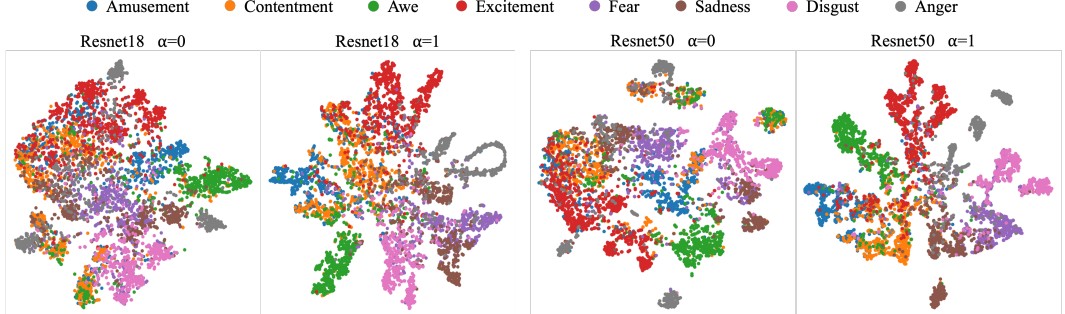

● Amusement ● Contentment ● Awe ● Excitement ● Fear ● Sadness ● Disgust ● Anger

Resnet18 α=0      Resnet18 α=1      Resnet50 α=0      Resnet50 α=1

Figure 4: In EmoSet, Eq. 14 is compared with the t-SNE graph [37] between alpha equals 0 and 1, where the left two pictures are the results of Resnet18 [13], and the right two pictures are the results of Resnet50.

In this way, we clearly explain the relationship between ACC and cross-entropy, and put forward the expectation of emotional label arrangement on the basis of emotional distance, so as to transform ACC and cross-entropy into ECC and ListMLE, that is, why ListMLE can be used as the backbone of ACC.

## B Experiment Analysis

Here we further explore the performance of ListMLE in visual emotion recognition tasks. The ListMLE loss function is designed to constrain the final prediction probability of the sample to conform to a given rank arrangement. According to the emotional distance in Eq. 1 defined in the measure, we convert it into the label order to cooperate with ListMLE for training. For example, if a image is labeled excitement, the defined labels are sorted from front to back as excitement,awe,contentment,amusement,fear,sadness,disgust and anger. We mix ListMLE with cross-entropy loss, where $\alpha$ is a hyperparameter to adjust the proportion of ListMLE.

$$\mathcal{L}_c = \mathcal{L}_{CE} + \alpha\mathcal{L}_{ListMLE}, \tag{14}$$

We compare ListMLE and cross-entropy loss on three backbone. The details are in Tab. 4. Similar to the conclusion of previous work [1], when focusing on the results of misclassification, accuracy will be reduced to some extent. When alpha is 1.0, because accuracy decreases more, ECC is worse than only using cross-entropy loss, but EMC performs best. When alpha is equal to 0.2, a better trade-off is reached between ACC and EMC, and ECC reaches the highest. In order to further emphasize the difference between ListMLE and cross-entropy, we present the visual comparison diagram in Fig. 4. It can be seen that the visualization result of ListMLE has a better clustering effect, and the relationship between categories is easier to distinguish, and it is more consistent with our defined emotional distance, indicating that ListMLE has learned the label structure information of emotion. Furthermore, this indicates a high correlation between our measures and tasks involving clustering, such as emotional image retrieval.

## C   Implementation Details

The above experiments Tab. 4 and Tab. 3 are carried out on three backbone, including ResNet18, ResNet50 and ResNet101 [13]. All models used pretrained weights in ImageNet [7] before training. For FI , all train images were resized to 256 * 256. To reduce overfitting, we randomly crop the image to 224 * 224 and flip it horizontally randomly. For test images, we also resize the image to 256 * 256, then make a 224*224 crop in the center of it. For EmoSet, we follow the experimental setup of the author [44] for the transform of the dataset. We use SGD with a momentum of 0.9 to optimize the network and we use a learning rate of 0.001. After 60 epochs, we decay the learning rate to 0.0002. Specifically, We warm up in the first epoch, which means the learning rate gradually increases to 0.001 in each iteration. The batch size of training data is 64. We are based on Pytorch for our experiments [27].

All experiments 4 3 1 are performed on two RTX 3090 GPUs. Each of these GPU has 24 GB of memory. For each set of experiments, it takes one to two days to use an RTX 3090.

## D   Limitation

 Finally, we would like to discuss the limitations of this work. For the threshold adjustment method in semi-supervised learning, we only introduce a simple and direct method. However, using our measures to guide the adjustment of thresholds in semi-supervised learning provides a novel perspective that deserves further investigation. Moreover, we should explore the application of our new measures in more fields, such as pre-training models and large language models, to fully validate the effectiveness of our measures.

