# OpenReview forum: "To Err Like Human: Affective Bias-Inspired Measures for Visual Emotion Recognition Evaluation"
_NeurIPS.cc/2024/Conference — NeurIPS 2024 poster_

### Official Review · Reviewer_jkV6 · 2024-07-10

**Soundness:** 2
**Presentation:** 3
**Contribution:** 2
**Rating:** 4
**Confidence:** 4

**Summary:**

The principal subject of this paper is the definition of the concept of misclassification in the field of visual emotion recognition, and the proposal of a novel evaluation method based on Mikel's Wheel distance to assess the degree of misclassification in methods of visual emotion classification. The paper also discusses the application of these evaluation methods in selecting models and adjusting label thresholds in semi-supervised learning tasks.

**Strengths:**

The primary contribution of this paper lies in the introduction of a novel emotional recognition evaluation framework that accounts for emotional similarity and the severity of misclassification. Furthermore, the paper applies this framework within the context of semi-supervised learning tasks.

**Weaknesses:**

1. In the related work section, subsections 2.2 and 2.3 appear to be erroneously treated as separate entities, whereas they should logically be combined into a single subsection, given their interconnected content.

2. Within the Emotion Wheel, adjacent emotions typically reflect interconnected affective experiences. However, the authors' adjustments to the numbering or distances on the Emotion Wheel, based on emotional polarity, have regrettably separated correlated emotion pairs such as excitement and fear, as well as amusement and anger, while paradoxically bringing the unrelated emotions of amusement and fear closer together. This configuration introduces substantial issues.

3. The assertions depicted in Figure 1 raise concerns; trends across various models, with the exception of the final MAPNET, demonstrate substantial consistency. Moreover, given the variations in model architectures, it is illogical to evaluate the differences in ACC and EMC performance based purely on release dates or ACC statistics.

4. In practical contexts, the emphasis on errors does not necessarily correlate with inaccuracies in polarity, suggesting that the EMC's utility may be limited.

5. While semi-supervised learning is utilized due to the difficulties in annotating data for emotion recognition tasks, the discussion of new evaluation metrics necessitates more accurate methodologies to effectively demonstrate these metrics' utility. Therefore, integrating experiments with fully supervised learning is advised.

6. In Section 4.2.1, the authors assert that models trained using CE loss achieve superior ACC performance, whereas those trained with a composite loss function, despite lower ACC, more effectively address misclassification and produce higher-quality pseudo-labels. Nevertheless, it remains ambiguous why subsequent sections claim an accuracy improvement of 1% over $S^2$-VER using our methodology.
7. Table 3's presentation lacks clarity. The meaning of "Our Rank" is unclear, and the justification for comparing RA w/o R1 with RE w/o R1 is not thoroughly elucidated.

**Questions:**

Same problems as mentioned above.

**Limitations:**

The authors mention some limitations in semi-supervised learning. However, it is recommended that they include a discussion of related work on supervised learning.

---

> ### Author Rebuttal · Authors · 2024-08-06
>
> ## 1 Weakness 2，7:
> As shown in Sec.3 of our rebuttal material，Emotional polarity is extremely important for emotion classification tasks. Therefore, we separate emotions of different polarities based on emotional polarity, but we do not shorten the distance between unrelated emotions. For example, before modification, the distance between amusement and fear was 4, and after increasing the distance, it became 8. To prove the effectiveness of our proposed changes, we followed previous work on misclassification[1] to verify the effectiveness of our proposed emotional distances. We converted these distances into ranking and trained with the mixed loss function based on different ranking.
>
> ## 2 Weakness3:
> Although the methods in Fig.1 of our manuscript are listed chronologically, the later methods are not necessarily developed based on the earlier ones. For example, when comparing OSANet and ResNet101, although OSANet’s accuracy is much higher than that of ResNet101 and ResNet50, its error severity is also higher. Similarly, even though WSCNet’s accuracy is comparable to ResNet50 and ResNet101, its error severity is very high. With this figure, we want to demonstrate that the current mainstream emotion classification methods do not incorporate the important factor of emotional error severity into the method design and evaluation system.
>
> ## 3 Weakness4:
> From the confusion matrix in Fig.1 of our rebuttal material, it can be seen that, in general, misclassification into categories with the same polarity occurs more often than into categories with opposite polarity. Taking OSANet as an example, although there is one more misclassification to emotion categories with opposite polarities (4 categories) than to those with the same polarities (3 categories), the proportion of misclassifications to categories with the same emotion polarity is 22.625%, while it is 13% for opposite polarities.This indicates that images with the same emotional polarity often express similar patterns and have relatively close emotions. This shows that there is a close relationship between emotional polarity and emotional misclassification. We also demonstrated through the validation experiment in Section 4.3 that the emotion rank, derived from the emotion distance we defined, has good properties.
>
>
> ## 4 Weakness5:
> We combined the experiments with fully supervised learning to analyze the emotion retrieval and binary classification tasks. As shown in the table, under the same model architecture, when the ACC is not significantly different, models with higher EMC, due to the incorporation of the concept of label distance, tend to perform better on eight retrieval metrics than models with higher ACC. In the binary classification task, models with higher EMC also show better ACC2 performance.
>
>
> ## 5 Weakness6:
> In semi-supervised training, the quality of pseudo-labels determines the final performance. Although models trained with CE loss have higher ACC, the pseudo-labels may be misclassified into emotion-irrelevant labels. For example, an “amusement” sample might be labeled with a pseudo-label of “fear,” which is disastrous for model training. On the other hand, while models trained with a combined function may have lower ACC, they have higher EMC. This means that although the pseudo-labels may be incorrect, they are likely to be mislabeled into categories close to the true category. Under this positive reinforcement, models with higher EMC are trained with continuously generated high-quality pseudo-labels, ultimately achieving higher ACC than the former.
>
> ## References
> [1].Bertinetto, Luca, et al. "Making better mistakes: Leveraging class hierarchies with deep networks." Proceedings of the IEEE/CVF conference on computer vision and pattern recognition. 2020.

---

> > ### Comment · Reviewer_jkV6 · 2024-08-09
> >
> > Thank you for the author's response. This addresses some of my concerns, hence I will increase my score.

---

> > > ### Author Response · Authors · 2024-08-09
> > >
> > > We are very glad to address your other concerns and add the corresponding experiments to the manuscripts later.

---

### Official Review · Reviewer_MhCQ · 2024-07-10

**Soundness:** 2
**Presentation:** 2
**Contribution:** 2
**Rating:** 5
**Confidence:** 3

**Summary:**

This paper proposes a novel evaluation approach based on Mikel’s emotion wheel from psychology, which considers the "emotional distance" between different emotions. It is claimed that the measure design considering the granularity of the emotions can be a better metric to evaluate visual emotion recognition. Experimental results adopting some of the semi-supervised learning methods and user studies show that the proposed metric is more effective than accuracy in evaluating performance and aligns with human emotional cognition.

**Strengths:**

- The first attempt to consider Emotion Granularity in the big datasets and propose a new evaluation procedure. The idea is innovative and can contribute but also see the weaknesses, especially in the emotion model considered as well as the comparisons.
- User study with 30 participants confirming the results are inline with human cognition.
-

**Weaknesses:**

-The use of Mikel's Wheel, a relatively less-studied emotion model in the context of building computational methods for automatic recognition, limits the study's impact. The first work utilizing this model dates back to 2016. Given the extensive body of work based on e.g., Ekman's model, the significance of this study is unclear.
- An important aspect completely overlooked in this paper is the frequent evaluation mechanism involving binary classification for each emotion class. Multiple emotions can be experienced simultaneously, making a multi-class classification task inappropriate. Consequently, accuracy (ACC) should be applied to individual emotion classes, where the positive class indicates the presence of a specific emotion and the negative class indicates its absence. Additionally, several methods have adopted the use of F1-macro and F1-weighted metrics, which were neither discussed nor applied in this study.
-Another issue that caught my attention is the limited range of researchers cited in the related work section regarding the use of Mikel's Wheel. Only the same group of researchers (e.g., Jingyuan Yang, Dongyu She) is cited, indicating that the impact of this work may be limited.
-The context is crucial in defining emotional distances, which seems to be missed in this study.
-References to CES and DES are missing (see Line 163).
-I expected to see more comprehensive comparisons in Tables 1 and 2. For example, comparisons could include works like "Li, J., Xiong, C., Hoi, S.C.: Comatch: Semi-supervised learning with contrastive graph regularization. In: ICCV (2021)." Currently, the comparisons are not as extensive as to confirm the claims.
- I do not find the venue for this submission appropriate. This work should be submitted to conferences focused on affective computing, such as ACII, where the community can discuss the theoretical underpinnings of the metric in conjunction with psychological claims and findings.

**Questions:**

- Have you considered using emotional lexicons that include a broader range of emotion classes?
-How were the results of S2Ver on Emoset produced?

**Limitations:**

- Emotion and overall affective computing always have a social impact but it is not discussed in this paper.
- Limitations are discussed in the appendix, while can be improved.

---

> ### Author Rebuttal · Authors · 2024-08-06
>
> ## Weakness:
> - First, we have added more extensive semi-supervised comparative experiments in Tab.1 of our rebuttal material, as well as experiments in Tab.2 of our rebuttal material on image retrieval tasks and emotion binary classification tasks.
>
> - Secondly, we strongly agree with your view that “human emotions are not singular,people experience a range of emotions.” Emotions have similarities, so accuracy in traditional classification tasks is not entirely applicable in emotion classification. Our metrics and the loss functions we provided incorporate this design. However, the task you mentioned is essentially a multi-label classification task, similar to emotion distribution learning. Both tasks face challenges in data annotation, and label quality may not always be reliable. It is difficult to determine which specific emotions are present in an image and how the emotional distribution of an image looks.In response to the first point of Reviewer 1, we also addressed this issue. Previous emotion distribution learning has often provided rough labels, and directly using KL loss to force the model’s final distribution to match the ground truth distribution is unreasonable. Therefore, we believe that constraining emotions through emotional ranking at the loss level is more appropriate. This approach helps the model learn which emotions are predominant in an image and which emotions are more prevalent relative to others.
>
> - Discussion of F1-macro and F1-weighted metrics: In fact, these are common metrics in machine learning for misclassification, but they are fundamentally different from our metrics. F1-macro is the average of the F1 scores for each category, while F1-weighted is the weighted sum of the F1 scores for each category. However, the precision and recall in F1 scores are calculated based on correct classifications, which differs from the research significance of our metrics. We aim to highlight the correlation of labels, where each category has a proximity relationship with other categories, but F1 scores cannot reflect this relationship.
>
> - However, we observed many works on facial expression recognition and error severity classification at NeurIPS. Our work could inspire thinking in these two fields and even broader areas. Therefore, we believe that submitting to NeurIPS is still meaningful.
>
> - Finally, we want to discuss the Mikel’s wheel and Ekman models. Any emotion model has varying distances between related emotional labels. Ekman does not have a mature distance definition like Mikel’s wheel, so we did not conduct experiments on Ekman. However, if a reasonable six-category distance for Ekman is defined, or if we define emotional distances based on Mikel’s wheel for the emotions in Ekman, the same metrics can be used. Additionally, Jingyuan Yang and Dongyu She are not a research group. Mikel’s wheel is the foundational model for the eight-category visual emotion model. Almost all visual emotion tasks are now conducted on FI, and further works like EmoGen and EmoEdit, which are based on the eight-category model, are built on this. Therefore, there is no need to worry about the limited scope of researchers using Mikel’s wheel.
>
> ## Question:
> - Yes, we have considered classification based on Plutchik’s model, which is a more fine-grained 24-category model built on the eight primary categories. If using such a model, similar to Ekman’s, it would only require defining a reasonable emotional distance model, such as assigning a weight to each layer of emotions in Plutchik’s model.
>
> - We processed EmoSet in the same way as FI because both are eight-category models based on Mikel’s wheel. Aside from the difference in dataset size, there is no significant difference between the two datasets during training. All our experiments on EmoSet used the same parameter settings as FI.

---

> > ### Comment · Reviewer_MhCQ · 2024-08-11
> >
> > Dear Authors,
> >
> > Thank you for your rebuttal. You have made an effort to address the implementation of your method with more widely recognized emotion models, such as those of Ekman and Plutchik. However, despite this concern being raised by another reviewer (Reviewer 133Y), your discussion remains inadequate. This reinforces my concern about the method's limited applicability, especially given its reliance on Mikel’s wheel, an emotion model that is one of the least utilized and referenced in the field.

---

> > > ### Author Response · Authors · 2024-08-12
> > >
> > > ## Response
> > > Thank you for your response.
> > >
> > > - First, we would like to say that the wheel model can also be extracted from Ekman and Plutchik to build our ECC and EMC, but we chose Mikel’s model as it already offers us an off-the-shelf wheel model.
> > >
> > >   For example, in Ekman’s model, there are four negative emotions that are the same as those in Mikel’s wheel. These emotion categories can directly use Mikel’s distances. For happiness and surprise, we only need to ensure that they are classified as positive emotions. The emotion distance matrix is defined as follows:
> > >
> > >   |Fear      | Sadness | Disgust| Anger|Happiness|Surprise|
> > >   | ------- | ------- |------- | ------- |------- | ------- |
> > >   | 1      | 2       |3      |4       |7       |6       |
> > >   | 2      | 1       |2      |3       |8       |7       |
> > >   | 3      | 2       |1      |2       |7       |8       |
> > >   | 4      | 3       |2      |1       |6       |7       |
> > >   | 7      | 8       |7      |6       |1       |2       |
> > >   | 6      | 7       |8      |7       |2       |1       |
> > >
> > >
> > > - Second, Mikel’s wheel is a model for eight categories of emotions, with the earliest paper[1] appearing in 2005, not 2016. Mikels et al.[1] through rigorous psychological research, categorizes emotions into eight types. Additionally, the paper points out that happiness, one of the six categories in Ekman’s model, often manifests as a mixed emotion rather than a single discrete emotion, which is clearly inappropriate.
> > >
> > >   Furthermore, a significant portion of research[2,3,4] based on art theory has defined eight different pixel-level features, which have been proven to be related to emotional responses. Therefore, the eight-category emotion model is a more reasonable classification model and has been extensively studied.
> > >
> > >   In addition, from the existing image emotion datasets, taking[5] as an example, there are 8 datasets using Mikel’wheel, but only 3 datasets using Ekman, and this does not include EmoSet.
> > >
> > >   So we think the Mikel's Wheel is widely used.
> > >
> > >
> > > ## References
> > > [1]Mikels, Joseph A., et al. "Emotional category data on images from the International Affective Picture System." Behavior research methods 37 (2005): 626-630.
> > >
> > > [2]Itten, Johannes. "The art of color: the subjective experience and objective rationale of color." (1961).
> > >
> > > [3]Valdez, Patricia, and Albert Mehrabian. "Effects of color on emotions." Journal of experimental psychology: General 123.4 (1994): 394.
> > >
> > > [4]Machajdik, Jana, and Allan Hanbury. "Affective image classification using features inspired by psychology and art theory." Proceedings of the 18th ACM international conference on Multimedia. 2010.
> > >
> > > [5]Zhao, Sicheng, et al. "Affective image content analysis: Two decades review and new perspectives." IEEE Transactions on Pattern Analysis and Machine Intelligence 44.10 (2021): 6729-6751.

---

> > > > ### Comment · Reviewer_MhCQ · 2024-08-12
> > > >
> > > > It seems that even when applying other emotion models, you still rely on Mikel's wheel. For instance, in the case of Ekman's model, categorizing emotions like happiness and surprise as strictly positive is overly simplistic and limiting. Moreover, directly comparing different emotion models is problematic (and not something typically done by psychologists) because they do not align perfectly—for example, surprise, which can vary greatly in valence.
> > > >
> > > > If I apply the same logic to using Mikel's wheel, which is associated with five datasets in [5], then it seems more appropriate to explore sentiment analysis, as it is linked to 11 datasets according to [5].
> > > >
> > > > Nevertheless, your paper presents a method validated only for Mikel's wheel, which I still find limited. [5] has shown that other datasets exist, meaning that other emotion theories could have been tested using your distance measure. Such an approach would significantly enhance the impact of the paper.

---

> > > > > ### Author Response · Authors · 2024-08-12
> > > > >
> > > > > ## Response
> > > > > Thank you for your response.
> > > > >
> > > > > We agree with your point that discussing emotion models is problematic, so we will not continue this discussion.
> > > > >
> > > > > What we want to address is the difference between object recognition and emotion recognition. In traditional object recognition, misclassification-related research is often based on WordNet, it can be used to define distances between categories. However, there is a fundamental difference between emotion recognition and object recognition. As we discussed earlier, there is no definitive answer as to whether six or eight discrete emotions better represent basic human emotions, whereas object recognition is developed based on ImageNet categories. This highlights a key difference between the fields.
> > > > >
> > > > > Returning to our paper, We very much hope that our metric can be applied to all emotion datasets, but this inevitably requires modeling an emotion distance calculation method for every emotion model. Mikel and Ekman already encompass most of the mainstream datasets and research in the current emotion recognition field. What we need is a comprehensive emotion category model to address this issue, but that is beyond the scope of this paper. Our primary goal is to reveal that accuracy is not a suitable metric for emotion analysis. Based on this, we propose a new evaluation metric grounded on the majority of existing work and provide corresponding baselines.
> > > > >
> > > > > Finally, we will follow your valuable review comments and add how the Ekman model uses our metrics to our future manuscripts.

---

> > > > > > ### Author Response · Authors · 2024-08-14
> > > > > >
> > > > > > Dear Reviewer,
> > > > > >
> > > > > > We have addressed the concerns you raised last time. We benefit a lot from the discussion with you, and we are looking forward to have more suggestions from you and continue the discussion.

---

### Official Review · Reviewer_133Y · 2024-07-11

**Soundness:** 3
**Presentation:** 2
**Contribution:** 3
**Rating:** 5
**Confidence:** 3

**Summary:**

The paper proposes a new measure for emotion recognition performance based on Mikel’s emotion wheel. The measure takes the distance between emotions into account. Experiments in semi-supervised learning on emotion recognition and user study demonstrate the effectiveness and superiority of the proposed metrics over accuracy.

**Strengths:**

1. Good performance compared with baselines in tables 1 and 2. Besides, standard deviations are reported which demonstrates the improvement is significant.
2. The idea of designing new measures for ER is novel and easy to follow. Though, it may not be applicable to the basic emotion space.

**Weaknesses:**

1. Few baselines are compared. Tables 1 and 2 are the main experiments while only one and two baselines are discussed and compared.
2. Section 4.2.1, the setting of pseudo labeling evaluation is confusing to me. How did you get the accuracy for models with different label num? Is there an official test set in FI/EmoSet for evaluation? Or did you use the whole set? If that's the case, what’s the difference between pseudo labeling evaluation and simple model training/testing evaluation?
3. Section 4.3 is also unclear to me. Can you clarify the annotation instruction in the user study? I am not sure how participants vote for models with higher ECC or ACC.
4. Mikel’s emotion distance is not applicable to Ekman six basic emotions which is widely used in most categorical emotion datasets.

**Questions:**

1. Line 304-307, how do you select the values for different hyperparameters?
2. How do you measure the standard deviations in tables 1 and 2?

**Limitations:**

Please refer to weaknesses

---

> ### Author Rebuttal · Authors · 2024-08-06
>
> ## 1 Weakness1:
> We have added comparative methods for the semi-supervised experiments in Tab.1 of our rebuttal material.
>
> ## 2 Weakness2:
> In fact, it is a semi-supervised learning experiment setting. We divided the training set according to a predetermined number of labeled samples into labeled training samples and samples we consider unlabeled. Then, we trained on the partially labeled training samples and a large number of unlabeled training samples and tested on the official test sets of FI/Emoset. During training, we can assess the model’s ability to label pseudo-labels by obtaining the true labels of the pseudo-labels. In the early stages of using different functions (CE and mixed functions), the model trained with CE loss has higher ACC for labeled pseudo-labels but lower EMC. In contrast, the model trained with mixed loss has lower ACC but higher EMC, which means the model can learn useful knowledge from incorrectly labeled tags that are similar to the true labels, thereby improving the final semi-supervised classification accuracy.
>
>
> ## 3 Weakness3:
> We added a description of the user study in Fig.2 of our rebuttal material. First, we selected samples that both models misclassified. Then, we presented the classification results of the two models, along with ‘Indistinct’ as three options, and asked users to choose which emotion was closer to the true emotion of the image
>
>
> ## 4 Weakness4:
> Our metrics primarily emphasize the similarity between emotional labels, as there are differences in the quality of misclassifications in emotions. However, Mikel’s wheel is off the shelf. Objectively speaking, Ekman also has emotional label similarities. It can be used as long as an appropriate model is defined just like Mikel’s wheel.
>
>
> ## 5 Question1：
> Based on existing semi-supervised work such as S2VER, FixMatch, and FlexMatch, we chose 0.98 and 0.7 as the upper and lower bounds of the threshold, and $\tau$ is 0.95. For  e , we derived empirical values for each dataset. For example, in the FI dataset, EMC is usually around 0.5, and in Emoset, EMC is usually around 0.6, so we set them to 0.5 and 0.4, respectively.
>
> ## 6 Question2:
> We followed the previous papers FlexMatch and S2VER, conducting the experiments three times under the same experimental settings with different random seeds, and then we calculated the standard deviation.

---

> > ### Comment · Reviewer_133Y · 2024-08-13
> >
> > Dear authors,
> >
> > Thanks for the comments and hard work. My major concerns have been addressed. I will raise my ratings and vote to accept the paper.

---

> > > ### Author Response · Authors · 2024-08-13
> > >
> > > Dear reviewer，
> > >
> > > We are very glad to address your other concerns and add the corresponding experiments to the manuscripts later.

---

### Official Review · Reviewer_Ro51 · 2024-07-12

**Soundness:** 2
**Presentation:** 2
**Contribution:** 2
**Rating:** 5
**Confidence:** 3

**Summary:**

This paper proposed new measures to evaluate the severity of misclassifications in visual emotion recognition. It addresses the limitations of traditional accuracy metrics by considering the psychological similarities between emotions. Utilizing Mikel's emotion wheel, the authors define an emotional distance metric and apply it to create a cost matrix for emotion classification. They introduce the Emotion Confusion Confidence (ECC) and Emotional Mistakes Confidence (EMC) metrics and validate their effectiveness through experimental results and user studies, demonstrating their robustness in semi-supervised learning contexts.

**Strengths:**

This work introduced the concept of mistake cost into visual emotion recognition, enhancing the evaluation of emotion classification methods by considering the psychological impact of misclassifications. By proposing new measures based on Mikel’s emotion wheel, it aids in model selection and threshold adjustment within semi-supervised learning, leading to improved classification performance. The study's validation through user studies confirms that the proposed metrics are consistent with human emotional cognition, offering a robust framework for future research.

**Weaknesses:**

1. The proposed Emotion Confusion Confidence (ECC) and Emotional Mistakes Confidence (EMC) metrics offer limited contribution as new evaluation standards to the community. Moreover, the exploration of these new metrics and their application in the manuscript is insufficient, resulting in a lack of depth and richness in the content.

2. The manuscript includes too few comparative methods, and the comparison forms and application richness are insufficient, which limits its credibility. In the User Study, only the analysis of ECC is included, with no analysis of EMC.

**Questions:**

1. The authors frequently mention the example: “misclassifying ‘excitement’ as ‘anger’ apparently is more severe than as ‘awe’.” How many methods currently misclassify this category of emotion as anger? Is there corresponding data to support this claim?

2. To fully demonstrate the effectiveness of the proposed new metrics, should these metrics be used with a more diverse set of methods? Additionally, the mainstream methods should be re-evaluated using the proposed metrics and compared with the accuracy (Acc) metric. Analyzing the strengths and weaknesses of methods from multiple dimensions could illustrate the effectiveness and advantages of the proposed evaluation metrics.

**Limitations:**

The paper acknowledges several limitations, including the use of a simple and direct method for threshold adjustment in semi-supervised learning. While the proposed measures offer a novel perspective, further investigation is needed to refine these methods. Additionally, the application of the new measures should be explored in other fields, such as pre-training models and large language models, to fully validate their effectiveness.

---

> ### Author Rebuttal · Authors · 2024-08-06
>
> ## 1 Weakness1: Contribution to visual emotion recognition
> - Emotional ambiguity and emotional relevance have always been important issues in the field of emotions, and previous work has been dedicated to solving these problems.Previous works[1] extended single labels into label distributions, or the work of Visual Sentiment Distributions[2][3], Whether it is based on the voted label distribution or the label distribution according to Gaussian fitting, only a rough and inaccurate distribution can be given, and there is a gap between ground truth label distribution and real-world label distribution. It is obviously unreasonable to give the same label distribution to all images of the same category according to Gaussian fitting, while ignoring the differences between the emotions of different images. Conversely, our measure using label rank information can make the ground truth label rank closer to the real-world label rank, or exactly equal, to better guide the model to learn the emotion categories structure.
> - From the perspective of misclassification, emotions have a natural adaptability compared to previous work. There is a clear correlation between different emotions, and there is no need for complex category distance definitions, which is of great significance for the field of visual emotion recognition.
>
>
> ## 2 Weakness2:
> As shown in Tab.1 and Tab.2 of our rebuttal material. We conducted supplementary experiments in semi-supervised settings and also tested our metrics on image retrieval and binary classification tasks. In the user study, we primarily analyzed ECC, which is equivalent to analyzing ACC+EMC. Previous work on misclassification has always aimed to reduce the severity of misclassification while ensuring the accuracy of ACC. Indeed, both ACC and EMC are generally used together to evaluate the model.
>
> ## 3 Question1：
> From the confusion matrix in Fig.1 of our rebuttal material, we can see that a significant portion of emotions are misclassified into categories with opposite polarity. For example, using OSANet, when the emotional polarity is positive, an average of 6.75% of the samples are misclassified as negative emotions. When the emotional polarity is negative, an average of 19.25% of the samples are misclassified as positive emotions.
>
> ## 4 Question2:
> In fact, we reassessed the current mainstream methods in Figure 1 of our manuscripts, where the relationship between ACC and EMC is shown.
>
> ## References
> [1].Yang, Jufeng, et al. "Weakly supervised coupled networks for visual sentiment analysis." Proceedings of the IEEE conference on computer vision and pattern recognition. 2018.
>
> [2].Yang, Jufeng, Dongyu She, and Ming Sun. "Joint Image Emotion Classification and Distribution Learning via Deep Convolutional Neural Network." IJCAI. 2017.
>
> [3].Yang, Jufeng, Ming Sun, and Xiaoxiao Sun. "Learning visual sentiment distributions via augmented conditional probability neural network." Proceedings of the AAAI Conference on Artificial Intelligence. Vol. 31. No. 1. 2017.

---

> > ### Author Response · Authors · 2024-08-14
> >
> > Dear Reviewer,
> >
> > We conducted some additional experiments and provided responses to your concerns. We sincerely hope to continue our discussion with you and are very keen to explore further insights and suggestions you might have.

---

> > > ### Comment · Reviewer_Ro51 · 2024-08-14
> > >
> > > Thank you for the author's efforts and responses, which address some of my concerns. After considering the other reviewers' comments and the rebuttal, I will raise my score.

---

### Author Rebuttal · Authors · 2024-08-06

## 1 Semi-supervised learning supplementary experiment
We added comparison methods for semi-supervised experiments, including CoMatch[1], SimMatch[2], SoftMatch[3] and FreeMatch[4]. Additionally, we combined the two tables into a new one. Please note that comparing other methods with our method in section 4.2.2 is unfair, as our method only involves adjusting the threshold. Nevertheless, our threshold-adjusting method is still better than some methods. Furthermore, the method in section 4.2.1 performs the best among all methods.


## 2 Full supervised learning supplementary experiment
We have demonstrated the various good properties included in our metric through fully supervised experiments. After training on base models (ResNet18, ResNet50, ResNet101) using CE loss and a mixed loss based on the metric, we evaluated the models on image retrieval and binary classification tasks. The experimental results show that models with higher EMC, meaning models with good misclassification properties, generally perform better in image retrieval and binary classification tasks.
- Image retrieval is essentially a type of metric learning. Boudiaf et al.[5] demonstrated that the final prediction score for each image category in metric learning actually reflects the mutual information between image features and labels. Yao et al.[6] experimentally verified that class-ordered pair-wise loss is effective for emotion retrieval tasks. Our metric aims to emphasize the spatial semantic structure of emotion labels, which essentially involves constraining the relationships between the prediction scores of each category, thus constraining the mutual information between image features and labels. This is why our metric can better distinguish the distances between different image features in image retrieval tasks.
- Furthermore, ACC2 is an important metric in the field of emotions, usually reflecting the model's ability in emotion binary classification (cite papers using ACC2). ACC2 actually corresponds to the classification results of the upper left ([0,3],[0,3]) matrix and the lower right ([4,8],[4,8]) matrix of the confusion matrix in our paper. Our metric makes more detailed considerations based on ACC2 and ACC.

In conclusion, usually, when the ACC is similar, a higher EMC indicates better model performance.

## 3 Explanation of the Importance of Emotional Polarity
Additionally, we illustrate the importance of emotional polarity by analyzing the classification confusion matrices of three mainstream methods (OSANet, SimEmo, MSPNet). The confusion matrices show that, in most cases, misclassifications tend to occur within categories with the same emotional polarity. However, some categories are also misclassified into those with opposite polarity. A considerable number of studies, such as S2VER, EASE[7] and TSL[8], address issues in the emotional domain through emotional polarity. This indicates that emotional polarity is a crucial feature of emotions, both from the perspective of human cognition and experimental results.

## 4 User Study
We provided a schematic diagram of the User study process. We selected images that were misclassified by both models, then created a questionnaire with the two misclassification results and "Indistinct", asking users to choose “which emotion is closer to the picture’s emotion?”

## References
[1].Li, Junnan, Caiming Xiong, and Steven CH Hoi. "Comatch: Semi-supervised learning with contrastive graph regularization." Proceedings of the IEEE/CVF international conference on computer vision. 2021.

[2].Zheng, Mingkai, et al. "Simmatch: Semi-supervised learning with similarity matching." Proceedings of the IEEE/CVF Conference on Computer Vision and Pattern Recognition. 2022.

[3].Chen, Hao, et al. "SoftMatch: Addressing the Quantity-Quality Tradeoff in Semi-supervised Learning." The Eleventh International Conference on Learning Representations.

[4].Wang, Yidong, et al. "FreeMatch: Self-adaptive Thresholding for Semi-supervised Learning." The Eleventh International Conference on Learning Representations. 2022.

[5].Bertinetto, Luca, et al. "Making better mistakes: Leveraging class hierarchies with deep networks." Proceedings of the IEEE/CVF conference on computer vision and pattern recognition. 2020.

[6].Yao, Xingxu, et al. "Adaptive deep metric learning for affective image retrieval and classification." IEEE Transactions on Multimedia 23 (2020): 1640-1653.

[7].Wang, Lijuan, et al. "Ease: Robust facial expression recognition via emotion ambiguity-sensitive cooperative networks." Proceedings of the 30th ACM international conference on multimedia. 2022.

[8].Zhang, Zhicheng, and Jufeng Yang. "Temporal sentiment localization: Listen and look in untrimmed videos." Proceedings of the 30th ACM International Conference on Multimedia. 2022.

---

### Decision · Program_Chairs · 2024-09-25

**Decision:**

Accept (poster)

**Comment:**

After the rebuttal period, the paper received three ratings of borderline accept and one of borderline reject.

The paper proposes to evaluate the performance in visual emotion recognition by considering the distance between different emotions on Mikel’s emotion wheel. Experimental results are conducted in semi-supervised learning on emotion recognition. The reviewers overall agree the approach is novel and innovative. Strengths of the paper include aiding in model selection and threshold adjustment with semi-supervised learning, validation through user studies, improved performance over baselines, and the novelty of new measure is good.

The original concerns, raised by the reviewers, include too few comparisons, confusing description of the evaluation and annotation, applicability of Mikel’s wheel to Ekman’s six basic emotions, multiple emotions occurring at once not being considered, separation of related emotions, and no fully supervised learning experiments.

The authors and reviewers took part in a good discussion during the rebuttal phase, with the authors providing significant clarification and new experimental results resulting in multiple reviewers increasing their scores. After the rebuttal phase, the paper strengths outweigh weaknesses as many of them were addressed. This paper can be accepted for publication.